# Effects of Genotype and Modified Atmosphere Packaging on the Quality of Fresh-Cut Melons

**DOI:** 10.3390/foods13020256

**Published:** 2024-01-13

**Authors:** Ranjeet Shinde, Yakov Vinokur, Elazar Fallik, Victor Rodov

**Affiliations:** 1Department of Postharvest Science, Agricultural Research Organization (ARO), The Volcani Institute, Rishon LeZion 7505101, Israel; rshinde@uoguelph.ca (R.S.); yvinokur@agri.gov.il (Y.V.); efallik@agri.gov.il (E.F.); 2The Robert H. Smith Faculty of Agriculture, Food and Environment, The Hebrew University of Jerusalem, Rehovot 7610001, Israel

**Keywords:** *Cucumis melo*, cultivar groups, ready-to-eat, shelf life, MAP, package perforation, headspace, fermentation volatiles, ethyl acetate

## Abstract

Marketing melons (*Cucumis melo*) as convenient fresh-cut products is popular nowadays. However, damage inflicted by fresh-cut processing results in fast quality degradation and food safety risks. The life of fresh-cut produce can be extended by a modified atmosphere (MA), either generated in a package by tissue respiration (a passive MA) or injected by gas flushing (an active MA). This work investigated the effect of passive and active MA formed in packages of different perforation levels on the quality of fresh-cut melons of two genetic groups: *C. melo var. cantalupensis*, characterized by climacteric fruit behavior, and non-climacteric *C. melo inodorus*. The best product preservation was achieved in passive MA packages: non-perforated for *inodorus* melons and micro-perforated for *cantalupensis* ones. The optimal packages allowed for the preservation of both genotypes for 14 days at 6–8 °C. The major factors limiting the shelf life of fresh-cut melons were microbial spoilage, translucency disorder and hypoxic fermentation associated with *cantalupensis* melons with enhanced ethyl acetate accumulation. *Inodorus* melons were found to be preferable for fresh-cut processing since they were less prone to fermented off-flavor development.

## 1. Introduction

Melons (*Cucumis melo* L.) are popular with consumers due to their high nutritional value, pleasant flavor and texture [1]. At the same time, melon is relatively inconvenient for consumption due to its large size and the presence of a considerable inedible part [2]. Therefore, marketing melons as convenient ready-to-eat sliced or cubed fresh-cut products is an attractive option for both consumers and retailers.

However, fresh-cut processing renders the fruit highly perishable, promoting physiological deterioration and providing a favorable environment for spoilage microorganisms and human pathogens [3]. In fresh-cut melons, cutting induces changes in texture, firmness and color that may result in a water-soaked appearance (translucency) of the tissues [4,5]. It affects respiration and ethylene production rates [6], as well as the profile of aroma volatiles [7,8]. Selecting the proper raw material in terms of genotype [9,10,11] and maturity [12] is important for ensuring the sufficient shelf life of fresh-cut melons. Product preparation should include efficient precut decontamination [11,13], aseptic processing with sharp tools [14,15], the storage of the final products under appropriate temperatures [16,17] and atmosphere composition [5,18].

A modified atmosphere (MA) with reduced oxygen and enhanced carbon dioxide levels can slow down produce deterioration via reduced respiration, inhibited ethylene biosynthesis and/or action and microbiostatic activity [19,20]. In traditional equilibrium (so-called “passive”) systems, the MA is generated due to produce respiration within semipermeable plastic modified atmosphere packaging (MAP). Such “passive” MA buildup may take a few days (a transient period) when the produce is exposed to non-optimal conditions and keeps deteriorating. In addition, a misbalance between high produce respiratory activity and insufficient package gas permeability may result in oxygen depletion and off-flavor development associated with the accumulation of fermentation volatiles, e.g., ethanol, acetaldehyde and ethyl acetate [21,22]. A micro-perforated MAP can prevent oxygen depletion by ensuring sufficient gas exchange, but it expands the transient period even more. On the other hand, in the case of “active” MAP systems, the package is flushed with an optimal gas mixture, diminishing the transient period. However, it does not eliminate and even aggravates the risk of hypoxia in the packages of highly respiring produce. The approach of micro-perforated active modified atmosphere (MAMA) packaging combines an active MA with the use of laser-micro-perforated plastic films in order to minimize the transient period and at the same time prevent hypoxia [23,24].

*Cucumis melo* is very diverse in fruit characteristics such as size, shape, color, texture, taste, composition and physiological behavior [25]. Based on fruit traits and uses, the species includes six cultivar groups, two of them being the most important commercially [2,26]. *C. melo var. cantalupensis* is characterized by medium–large-sized climacteric fruit with a netted, smooth or scaly rind of variable color. The fruits are aromatic, have sweet, juicy flesh and easily detach from the vine (“slip”) at maturity. The group includes dessert melon types such as Galia, Ananas, Charentais, ‘American shipper’ cantaloupes (muskmelons), etc. In contrast, *C. melo var. inodorus* is characterized by large-sized melons with non-aromatic, non-climacteric and relatively long-stored fruit with a thick, smooth, warty or wrinkled rind. This group includes sweet dessert melons, such as Honeydew, Piel-de-Sapo and Casaba types. Each of these types is represented by numerous cultivars. For example, over sixty Galia-type cultivars are available in the market [27].

The major objective of the present study has been investigating the performance of different MA packaging types as a means to preserve the quality of fresh-cut melons of the *cantalupensis* and *inodorus* groups during a simulated shelf life.

## 2. Materials and Methods

### 2.1. Plant Material and Fruit Processing

Green-fleshed Galia-type melons (*C. melo var. cantalupensis*, cv. ‘Raanan’, HaZera Genetics, Israel) and pink-fleshed Piel-de-Sapo-type melons (*C. melo var. inodorus*, cv. ‘Sorbeto’, Catom Seeds, Israel) were harvested at commercial maturity from commercial plots in the northern Arava Valley, Israel. The melons were selected for uniformity and maturity based on ground color, appearance and firmness and brought to the ARO—The Volcani Institute—in an air-conditioned vehicle on the day of harvest. The fruits were pre-cooled and kept at 6 °C for about 2 days until the processing. The melons were aseptically processed at the pilot fresh-cut facility at the Department of Postharvest Science, ARO—The Volcani Institute, Israel. They were soaked in a sodium hypochlorite solution (100 ppm active chlorine) for 10 min, brushed within this solution for 2 min with a stiff plastic brush, rinsed with tap water and allowed to dry under the flow of sterile air. During processing, the fruits were cut with a sharp, cleaned knife to prepare melon flesh chunks of approximately 2.5 cm in size.

### 2.2. Packaging and Storage

Chunks from the same melon were randomly distributed between polyethylene terephthalate (CPET) trays of 171 × 127 × 50 mm dimensions (MCP Performance Plastics, Kibbutz Hama’apil, Israel), ca. 100 g per tray. The trays were sealed to obtain different package types. In the case of a passive MA, the trays were sealed without gas flushing and contained regular air as the initial atmosphere. In the case of an active MA, the trays, before sealing, were flushed with a gas mix of 5% O_2_, 10% CO_2_ and 85% N_2_. Typically, the initial in-package atmosphere contained, in that case, 6 kPa of O_2_ and 9 kPa of CO_2_. The heat-sealing of both the active and passive MA trays and the gas flushing of the active MA trays were carried out with an ILPRA Food Basic packaging machine (ILPRA, Vigevano, Italy). The trays were sealed with a 35 µm thick polyester-based medium-barrier laminate lidding film Topaz-335 (Plastopil, Kibbutz Hazorea, Israel) of the following perforation levels: (a) non-perforated, (b) with one laser micro-hole of ca. 70 µm per tray, or (c) with two micro-holes of the same size per tray. To obtain pinhole-perforated packages (d), one thin-needle (0.5 mm) hole was created manually in the active-MA trays sealed with a non-perforated lidding film. In (e) the control macro-perforated packages, MA generation was prevented by the manual perforation of two opposite walls of the sealed trays (air as the initial atmosphere) using a hot instrument, 2.5 mm in diameter, altogether creating two holes per tray. Typically, at least ten replicate packages were prepared per treatment, each from a different melon. All packages were stored under simulated cooled shelf-life conditions (temperature of 7 ± 1 °C, relative humidity of ca. 90%). The packages were sampled on day 0 and after one and two weeks of storage. With *inodorus* melons, the amount of material allowed an additional sampling point after 12 days of storage.

### 2.3. In-Package Atmosphere Composition Analysis

The atmosphere composition inside the packages was measured for O_2_ and CO_2_ by using an OXYBABY 6.0 gas analyzer (WITT-GASETECHNIK GmbH & Co. KG, Witten, Germany) comprising an infrared sensor for CO_2_ measurements and an electrochemical sensor for O_2_. The instrument’s needle was inserted into the packages through adhesive rubber septa attached to the lidding film.

Headspace atmosphere samples of 8 mL were withdrawn from the packages via the above-mentioned septa using 10 mL gas-tight syringes with hypodermic 25 G needles (0.5 mm × 16 mm). The concentrations of acetaldehyde, ethanol and ethyl acetate were analyzed simultaneously with a Varian 3300 gas chromatograph (Varian, Inc., Palo Alto, CA, USA) equipped with a flame ionization detector and a 20% Carbowax 20 M packed column using helium as the carrier gas. The column, injector and detector temperatures were 80, 110 and 180 °C, respectively, as described by Poverenov et al. [28]. The concentration of ethylene was measured as described by Freiman et al. [29] by using a Varian 3300 GC instrument with a flame-ionization detector and a stainless steel column (length: 1.5 m; outside diameter: 3.17 mm; internal diameter: 2.16 mm) packed with HayeSep T, with a particle size of 0.125–0.149 mm (Alltech Associates, Inc., Deerfield, IL, USA). Helium was used as the carrier gas (5 mL/min).

### 2.4. Quality Assessment

The product quality was evaluated visually and organoleptically by three expert panelists (including one representing the industry) according to a scoring method as described by Van Oirschot and Tomlins [30]. For each quality parameter, the scores were assigned by a consensus decision of the panel according to the evaluation form received from the industry. The evaluated quality parameters included off-odor (severe to no), off-flavor (severe to no), piece shape (misshapen to clean-cut), translucency, i.e., water-soaked appearance (severe to no), decay, i.e., external signs of microbial spoilage (severe to no) and texture (mushy to crunchy). For all quality parameters, a uniform 5-grade scale was used, in which the scores of 1, 2.5 and 5 were, respectively, the worst, the marketability threshold and the best quality grade. The general quality decline was determined by the lowest score received by a sample in any category evaluated, i.e., a sample that obtained a score below 2.5 in any category was judged non-marketable. At least three typical packages were evaluated at each sampling point for each packaging type, and their scores were used as replications.

First, the panelists partially peeled off the plastic film and evaluated the off-odor intensity. Second, visual evaluations were conducted for piece shape, color, translucency and decay. In addition, the drip loss was measured with a pipette. At last, if samples were visually acceptable, the panelists performed off-flavor and texture evaluations. The panelists used water between samples to cleanse their palates. The experimental samples were presented in random order.

The soluble solids content (SSC, %) of the juice was determined with a digital refractometer (Atago Co. Ltd., 3210 Huncho, Itabashi Ku, Tokyo, Japan). The analysis was performed in triplicate using, as replication, a juice sample obtained by squeezing three pooled melon chunks in a cloth pouch. The flesh firmness (N) was measured with a penetrometer (Chatillon Digital Force Guage, New York, NY, USA) equipped with a 6 mm conical probe on three cubes from each replication tray, for a total of nine measurements per sampling point. The probe was inserted into the side surfaces of the melon chunk to omit the effect of the natural firmness gradient within the melon when the skin-facing side is firmer than the seed-cavity-facing one.

### 2.5. Statistical Analysis

The experiments were performed in triplicate and repeated at least twice for each melon cultivar. The results of a typical trial are presented in this paper. Microsoft Office Excel spreadsheets were used to calculate the means, standard deviations and 95% t-based confidence intervals. The statistical analyses used JMP Version 5.0.1 software (SAS Institute, 2003, Cary, NC, USA). The significant differences among the sample means were evaluated by a one-way analysis of variance (ANOVA) and, where appropriate, the means were differentiated post hoc by the Tukey honestly significant difference (HSD) test.

## 3. Results

### 3.1. Headspace Atmosphere Composition

#### 3.1.1. Oxygen

Similar oxygen dynamics were observed in the packages of the two melon genotypes. Package perforation was the major factor determining the oxygen level. In the non-perforated packages, complete oxygen depletion was reached in the active MA packages after one week of storage, and in the passive MA packages, after two weeks, i.e., at the end of the trial (Figure 1A,B). In the micro-perforated passive and active MA packages, the O_2_ steady-state concentrations varied between 15 and 18 kPa, depending on the perforation level. There was no change in the oxygen concentration of the macro-perforated control packages.

#### 3.1.2. Carbon Dioxide

The CO_2_ accumulation in the non-perforated active MA packages steadily increased and exceeded 21 kPa, indicating hypoxic fermentation [20]. Although the CO_2_ dynamics were generally similar in the packages of the two genotypes, the final CO_2_ level in these packages was slightly higher for the *cantalupensis* melons than for the *inodorus* ones, 25 vs. 22 kPa, respectively. On the other hand, in the non-perforated passive MA packages of both genotypes, the CO_2_ level stabilized at 17 kPa (Figure 1C,D). The steady-state CO_2_ ranges were 2.5–3, 5–6 and 7–8 kPa for pinhole, two-micro-hole and one-micro-hole perforated packages, respectively, both in the passive and active MAs (Figure 1C,D). Practically no CO_2_ accumulation was observed in the macro-perforated packages.

#### 3.1.3. Ethylene

In contrast to oxygen and carbon dioxide, the headspace ethylene concentrations differed markedly between the two genotypes, being approximately one order of magnitude higher in the packages of the climacteric *cantalupensis* melons than in those of the non-climacteric *inodorus* ones. Interestingly, the ethylene level was strongly affected by the MA type. The most drastic contrast was observed in the non-perforated packages, where the highest ethylene accumulation was observed in the non-perforated passive MA packages (0.5 and 5 ppm for the *inodorus* and *cantalupensis* genotypes, respectively), while the non-perforated active MA packages contained negligible ethylene levels (Figure 1E,F). Irrespective of perforation, the accumulation of ethylene in all the active MA packages did not exceed 0.1 ppm for the *inodorus* melons and 1 ppm for the *cantalupensis* ones. In the passive MA packages with one micro-hole, ethylene actively accumulated during the first week of storage but subsequently declined (Figure 1E,F). Apparently, the headspace ethylene level was affected by the interplay between the tissue biosynthetic activity suppressed by CO_2_ and ethylene diffusion from the packages to the outside atmosphere, depending on the film perforation.

#### 3.1.4. Fermentation Volatiles

In the samples of *inodorus* melon, during the first week of storage, there was no significant difference in the content of fermentative volatiles, especially in the passive MA packages, where similar results were observed in the non-perforated and macro-perforated packages (Figure 2A). At the same time, in the active MA packages, the least perforated packages (zero or one micro-hole) showed a trend toward somewhat higher ethanol levels. During the second week, the patterns of fermentation volatiles in the *inodorus* packages stabilized, showing significantly a higher accumulation of fermentation volatiles in the non-perforated packages than in all the perforated ones. Interestingly, ethanol was the most prevalent volatile in the non-perforated active MA packages, while acetaldehyde prevailed in the non-perforated passive MA packages. At the same time, in the perforated packages, the accumulation of fermentation volatiles in the active MA was somewhat higher than in the passive MA, mainly due to the enhanced acetaldehyde levels (Figure 2A). Ethyl acetate was observed in small amounts in the *inodorus* passive MA packages but was practically absent in the active MA packages.

The amount of fermentation volatiles in the headspace of the MA-packaged *cantalupensis* melons was typically 5–10 times greater than that of the *inodorus* ones. This difference was primarily due to the accumulation of ethyl acetate that prevailed in the headspace of all the MA-packaged *cantalupensis* melon samples but was just a minor ingredient in *inodorus* volatiles (Figure 2A,B). On the other hand, in the absence of MA, the macro-perforated packages of both genotypes showed relatively little ethyl acetate and similar total-level fermentation volatiles (ca. 5–10 ppm). The largest concentrations of fermentation volatiles were produced by the *cantalupensis* fresh-cut melons in the non-perforated active MA packages, ca. 80 and 150 ppm after 7 and 14 days of storage, respectively. In addition to ethyl acetate, these packages accumulated enhanced amounts of ethanol (Figure 2B). Interestingly, the *cantalupensis* passive MA packages with two micro-holes contained, after 14 days of storage, much more ethyl acetate vapor than those with one micro-hole.

### 3.2. Quality: Inodorus Melons

No decay was observed in the fresh-cut *inodorus* melons during the first 7 days of storage. Later on, the macro-perforated packages showed the most obvious decay development (Figure 3A), accompanied by a certain piece shape loss (Figure 3B). On the other hand, the non-perforated passive and active MA packages had no decay throughout the whole storage period. In all the perforated packages, the severity of decay was aggravated after 12 days of storage, except for the active MA package with 1 micro-hole, where the decay was minimal (Figure 3A).

The passive MA packages (non-perforated or single-micro-hole-perforated) showed the best odor (Figure 3C) and taste (Figure 3D) stability of the fresh-cut *inodorus* melons. Certain fermented off-odor and off-flavor (Figure 3C,D) notes were sensible in the non-perforated active MA packages and, during the last days of storage, in the active MA packaged with a single micro-hole. At the same time, a moldy off-odor was detected in the macro-perforated and pinhole-perforated packages. However, it should be noted that the off-odor and off-flavor scores in the tested *inodorus* melon samples never reached prohibitively strong scores and never declined below the marketability threshold of 2.5. The samples with visible decay were not tasted, so there were not enough replications for a statistical analysis of the off-flavor severity at a 15-day time point (Figure 3D).

Figure 3E clearly demonstrates that the active MA enhanced the translucency. This effect was especially evident in the non-perforated packages, although the difference in translucency between the various active MA treatments was not statistically significant. In addition, at the end of storage, an enhancement in translucency (water-soaked appearance) was observed in the macro-perforated packages as one of the decay manifestations. Practically no translucency was detected in the non-perforated passive MA packages (Figure 3E). The firmness and SSC levels in the fresh-cut *inodorus* melons were within 7–9 N and 9–11% ranges, respectively. No significant changes were detected during the storage of healthy melon pieces (Appendix A). However, the decay obviously resulted in tissue maceration and softening.

An integral characteristic of a product’s quality decline during storage is presented in Figure 3F. The fastest and most severe deterioration was observed in the macro-perforated packages containing no MA, primarily due to the decay development. The best quality preservation was ensured by the non-perforated passive MA. While the non-perforated active MA also showed good decay control, its positive effect was jeopardized by translucency aggravation.

### 3.3. Quality: Cantalupensis Melons

The trends in the microbial decay of the fresh-cut *cantalupensis* melons were similar to those observed with the *inodorus* type. The greatest microbial spoilage was evident in the macro-perforated packages containing no MA (Figure 4). During the second week of storage, mold-caused decay developed in the packages less protected by MA, i.e., the packages that were macro-perforated, pinhole-perforated and micro-perforated with two holes (Figure 5A). At similar perforation levels, the active MA packages showed lower decay severity than the passive MA ones. This decay was associated with a slight piece deformation (Figure 5B). On the other hand, the non-perforated or single-hole micro-perforated packages had no or negligible decay (Figure 5A).

In contrast to the *inodorus* samples, the fresh-cut *cantalupensis* melons had a distinct smell, combining the typical melon aroma with that or another degree of off-odor. A moldy off-odor was sensible during the second week of storage in the headspace of the macro-perforated packages, characterized by severe decay (Figure 4). Noticeable fermented off-odor and off-flavor appeared in the non-perforated active MA packages and, to a lesser extent, in the non-perforated passive MA and active MA packages with a single micro-hole (Figure 5C,D). Off-flavor was also registered in the active MA packages with two micro-holes (Figure 5D). Similar to the *inodorus* melons, the samples showing visible decay during the second week of storage (see Figure 5A) were not tasted at the 14-day time point. Therefore, the only treatment that showed acceptable edible quality throughout the storage period was passive MA with a single micro-hole (Figure 5D).

The translucency in the *cantalupensis* melons was less evident than in the *inodorus* ones, except for the macro-perforated packages, where water soaking was one of the signs of decay development. At similar perforation levels, the active MA packages had somewhat higher translucency scores than the passive MA ones, although the difference was usually statistically insignificant (Figure 5E). The firmness and SSC levels of the *cantalupensis* melons were within the ranges of 6–8 N and 9–10%, respectively. These values did not change significantly during storage, provided the fresh-cut pieces remained free from microbial decay (Appendix A).

Figure 5F summarizes all the deterioration phenomena in the fresh-cut *cantalupensis* melons as an integral general quality parameter. The most severe quality degradation took place in the macro-perforated packages due to the decay development and in the non-perforated active MA packages due to the prohibitive off-flavor and off-odor. On the other hand, the passive MA packages with a single micro-hole allowed for the best quality preservation of the fresh-cut *cantalupensis* melons during the 14-day storage period (Figure 5F). In a separate trial, we found that such packaging could extend the shelf life of fresh-cut *cantalupensis* melons up to 21 days.

## 4. Discussion

Modified atmosphere packaging (MAP) is defined as ‘the packaging of a perishable product in an atmosphere which has been modified so that its composition is other than that of air’ [31,32], and the resulting environment minimizes the physiological and microbial deterioration of the foods [33]. With respiring foods such as fresh produce, atmosphere modification is based on restricting gas exchange between the package interior and the environment [34]. The MA composition is determined by the balance between food’s respiration and gas diffusion through a package [33]. However, different commodities require diverse atmospheric compositions for their best preservation, and the barrier properties of available plastic films are not always suitable for reaching these conditions [35]. In particular, packaging highly respiring fresh-cut fruits and vegetables in existing plastic films may lead to undesirable hypoxic fermentation, and therefore perforation is required to avoid this condition. At the same time, excessive perforation may preclude MA formation, making the in-package atmosphere identical to the surrounding air [36]. Using micro-porous or micro-perforated packaging materials helps modulate the package barrier properties in order to create a desirable atmosphere [34,37,38]. To reach this purpose, the perforation size should not exceed 100–500 μm, and the perforation level should be optimized, preferably based on mathematical modeling [36]. Perforation-mediated MAP serves, nowadays, as the basis for modern industrially applied technologies for the preservation of fresh produce [39,40].

A modified atmosphere in a package can be created either solely by the respiratory activity of the product (so-called “passive MAP”) or actively (“active MAP”) by displacing air with a desired gas mixture and/or by using additives that absorb or release gases or volatile compounds [33,41]. In both the “passive” and “active” MAP versions, the equilibrium steady-state concentrations are determined by the balance between the respiration rate of the product and the diffusion characteristics of the packaging material. However, flushing a package with a gas mix can shorten the time needed to attain a desirable MA that may be critical for the preservation of products highly sensitive to oxygen or with a low respiration rate [33]. Nowadays, active MAP is a postharvest technology commonly applied to maintain the quality and extend the shelf life of fresh produce [41]. At first glance, the idea of combining the active MAP technique with micro-perforated packaging (micro-perforated active modified atmosphere, or MAMA packaging) seems paradoxical because it allows for a partial escape of the injected gas mix through perforations. However, it in face allows the MAP to be stabilized without a risk of hypoxia [24]. The positive effects of micro-perforated active MAP on produce preservation have been demonstrated with fresh-cut strawberries [23], cabbage [42], litchi [43] and rocket leaves [44].

The results of this work have confirmed the efficacy of MAs for the preservation of fresh-cut *cantalupensis* and *inodorus* melons [5,45]. The optimal MA solutions adjusted in our trials for each one of the two genotypes tested allowed their quality to be maintained for up to 14 days. Microbial decay and physiological disorders (fermentation and translucency development) were the major factors limiting the shelf life of the produce [5,9]. The major MA advantage was related to the control of microbial spoilage, primarily due to the fungistatic effect of the elevated CO_2_ levels [19,20].

At the same time, using a low-oxygen MA was associated with the risk of hypoxic fermentation due to eventually passing the anaerobic compensation point (ACP), resulting in flavor deterioration [21,22]. The two genotypes differed markedly in the amount and composition of volatiles generated under low-oxygen conditions. Ethyl acetate prevailed in the fermentation volatiles produced by the fresh-cut *cantalupensis* melons exposed to hypoxia. Ethyl acetate is a characteristic melon fermentation marker associated with an unpleasant solvent-like off-flavor [22,46]. The *inodorus* melons produced very little ethyl acetate, while their ethanol and acetaldehyde levels were comparable (typically, 1.5–2 times lower) with those of the *cantalupensis* packages. The difference in ethyl acetate production was most probably related to the genetically determined low expression of genes responsible for alcohol acetyl transferase (AAT) enzymes in the non-climacteric *inodorus* melon types [47,48]. Due to this difference, the total accumulation of fermentation volatiles in the headspace of the oxygen-deficient MA packages of the *inodorus* melons was 5–10 times lower than in the *cantalupensis* melons. Therefore, the fresh-cut *inodorus* melons could benefit from the decay control provided by non-perforated MA with a low risk of prohibitive off-flavor. Altogether, *inodorus* melon varieties may be preferable for fresh-cut processing since they are less prone to fermented off-flavor development.

The effects of micro-perforated packaging on the storage of fresh-cut melons are poorly presented in the literature. Aguayo et al. [49] showed no advantage of micro-perforated polypropylene passive MA packages for the preservation of fresh-cut *inodorus*-type ‘Amarillo’ melons compared with non-perforated ones, as was also found for *inodorus* melons in our study. On the other hand, with the MA-packaged *cantalupensis* melons, micro-perforation was essential for keeping their sensory quality at an acceptable level under the given storage conditions. Moreover, choosing an appropriate perforation level was critical for the realization of the MA potential, as illustrated by the significant difference in decay severity between the packages bearing one vs. two micro-holes. A minimal micro-perforation level (a single 70 µm hole per 100 g package) was helpful for the efficient preservation of the *cantalupensis* melons in the passive MA packages, simultaneously controlling decay and off-flavor generation.

In this study, we compared passive and active MAs as a means of preserving fresh-cut melons of two genetic types: *inodorus* and *cantalupensis*. Furthermore, the performance of the novel approach of MAMA packaging combining active MA with micro-perforation [24] was tested with fresh-cut melons. The difference in the atmosphere composition between the active and passive MA packages was evident during the first week of storage, while during the second week, the packages of the same perforation level reached similar steady-state concentrations of oxygen and carbon dioxide, irrespective of their initial atmospheric compositions. This observation was in line with the model prediction that the steady-state O_2_ and CO_2_ levels in MA packages depend on the product’s respiration rate and the packaging material’s permeability, but not on the initial atmosphere [23,24]. On the other hand, the initial atmosphere determined the conditions during the transient period and its duration. Thus, the non-perforated active MA packages reached hypoxia after one week of storage, while in the non-perforated passive MA packages, it took twice as long and happened just at the end of storage. Understandably, the non-perforated active MA resulted in stronger fermentation and flavor deterioration than the passive one, although the differences were not always statistically significant.

The combination of active MA with micro-perforation (MAMA packaging) was supposed to prevent the risk of hypoxia and, at the same time, stabilize the MA composition within a desirable range, ensuring efficient spoilage control. Indeed, the active MA packages with a single micro-hole allowed for the maintenance of a steady 8–9 kPa CO_2_ level throughout the storage period, which is close to the recommendations [50] and sufficient to control the microbial decay of the product for two weeks. In addition, the active MA packaging, either perforated or not, inhibited ethylene production by both melon genotypes, most probably due to the effects of CO_2_ [51] and/or ethanol vapor [52] on ethylene biosynthesis. This ethylene inhibition might have a positive effect on product preservation because fresh-cut melon deterioration is associated with ethylene effects [53,54].

At the same time, in contrast to reports by Bai et al. [9,18], the active MA in our trials tended to aggravate translucency development in both genotypes, especially in the *inodorus* melons. This trend toward higher translucency was observed in all the active MA packages, including the pinhole-perforated ones that had a very moderate degree of atmosphere modification (2–3 kPa CO_2_ and 18–19 kPa O_2_). We suggest, therefore, that the translucency in this case might be enhanced not by a certain atmosphere composition but by the gas flushing procedure per se, which included vacuum application. Pressure fluctuations may cause translucency in fruit tissues [55]. Water soaking and translucency in fresh-cut melons and watermelons are associated with tissue disruption and cell wall degradation [56,57]. *Inodorus* Sorbeto cv. flesh might be especially susceptible to this disorder due to its crispy texture and high turgidity. If our suggestion is true, the translucency development in active MA packaging might be alleviated by packaging machinery performing gas flushing without vacuum application.

The MAMA packages with a single micro-hole ensured that the headspace oxygen level remained well above the ACP, about 15 kPa. Nevertheless, the *cantalupensis* melons kept in these packages demonstrated an enhanced accumulation of fermentation volatiles, in particular ethyl acetate, and noticeable off-flavor. This phenomenon might be a result of local hypoxia caused by tissue water-soaking, as shown for fruit watercore disorder [58,59], possibly in combination with an elevated headspace carbon dioxide level. Altogether, the performance of the active MA and, in particular, the MAMA packaging observed in this study did not justify their advantage over the passive MA for the preservation of fresh-cut melons, in contrast to the results obtained previously with topped strawberries [23]. Furthermore, the most successful packaging solutions revealed by this study were a non-perforated passive MA for the *inodorus*-type melon and a single-micro-hole passive MA for the *cantalupensis*-type melon. Additional advantages of these packaging solutions for the industry are their technical simplicity and relative cost efficiency. Naturally, the choice of packaging solutions should reflect the respiratory activity of the object affected by its physiological peculiarities and storage environment, in particular observing the cold chain conditions.

## 5. Conclusions

The study has confirmed the efficacy of MA packaging for the preservation of fresh-cut melons, primarily due to the control of microbial spoilage.

Fermented off-flavor associated with ethyl acetate accumulation was the major risk of using a low-oxygen MA with fresh-cut *cantalupensis* melons.

*Inodorus* melons were found to be preferable for fresh-cut processing since they were less prone to fermented off-flavor development.

The active MA tended to aggravate translucency development, especially in the *inodorus* melons, presumably due to the effect of the vacuum-driven gas flushing procedure on the flesh tissue integrity.

No advantages justifying the application of active MA packaging to fresh-cut melons as compared to a passive MA were observed in this study.

The non-perforated passive MA packaging was suitable for the preservation of the fresh-cut *inodorus* melons, while a minimal micro-perforation level was needed for the *cantalupensis* melons in order to preserve their sensory quality.

The optimal packaging methods allowed for the good quality preservation of both genotypes for 14 days of shelf life at 6–8 °C.

## Figures and Tables

**Figure 1 foods-13-00256-f001:**
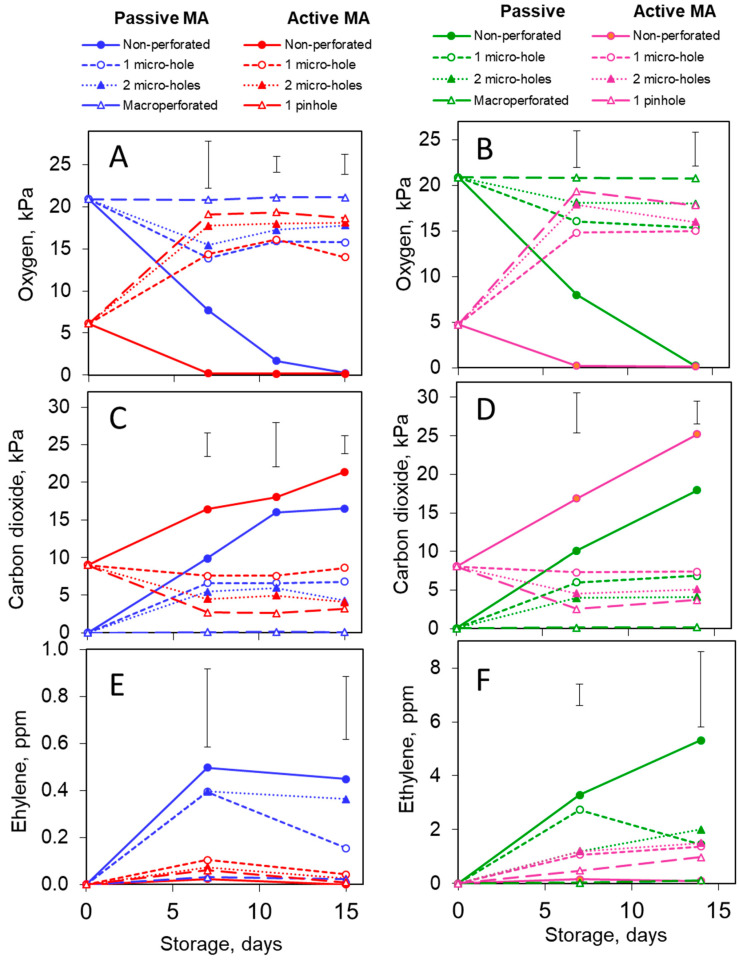
Effects of passive and active MA packaging on oxygen (**A**,**B**), carbon dioxide (**C**,**D**) and ethylene (**E**,**F**) levels in the headspace of packages containing 100 g of fresh-cut *inodorus* (**A**,**C**,**E**) or *cantalupensis* (**B**,**D**,**F**) melons during storage at 6–8 °C. Individual data points are the means of three replications. Vertical bars represent honest significant differences (HSDs) (*p* ≤ 0.05) for each sampling period determined by the Tukey HSD test.

**Figure 2 foods-13-00256-f002:**
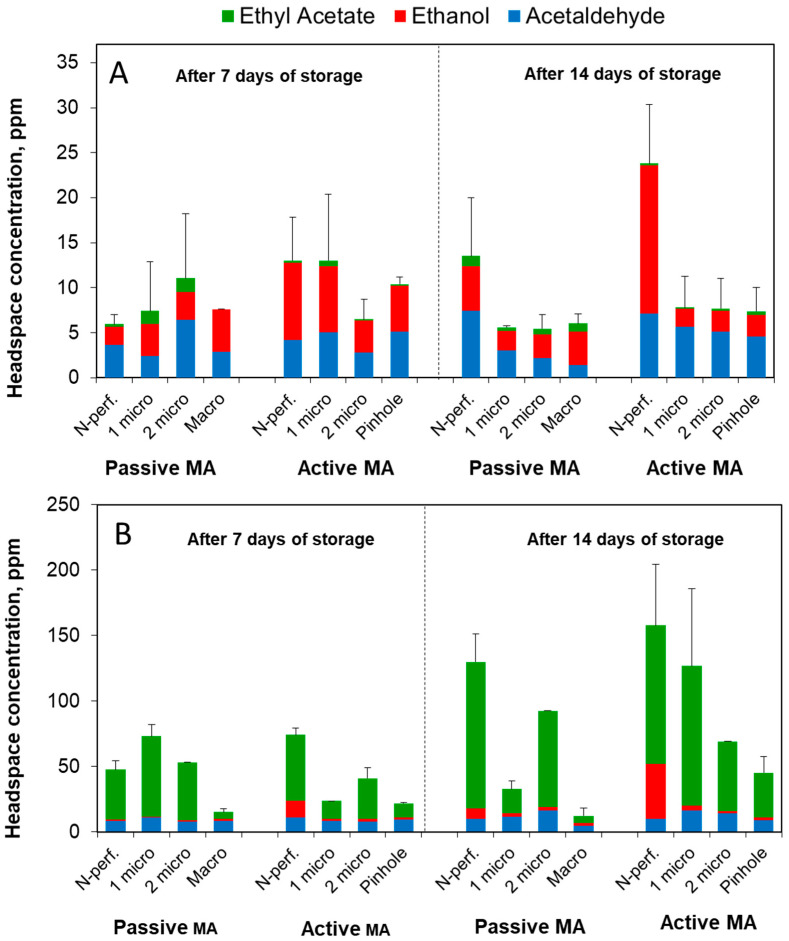
Effects of passive and active MA packaging on the content of fermentation volatiles (ppm) in the headspace of packages containing 100 g of fresh-cut *inodorus* (**A**) and *cantalupensis* (**B**) melons after one and two weeks of storage at 6–8 °C. Package perforation levels: non-perforated (N-perf.), a single 70 µm micro-hole (1 micro), two 70 µm micro-holes (2 micro), a single 0.5 mm hole (Pinhole), two 2.5 mm holes (Macro). Individual data points are the means of three replications. Vertical bars represent 95% t-based confidence intervals in the total content of fermentation volatiles.

**Figure 3 foods-13-00256-f003:**
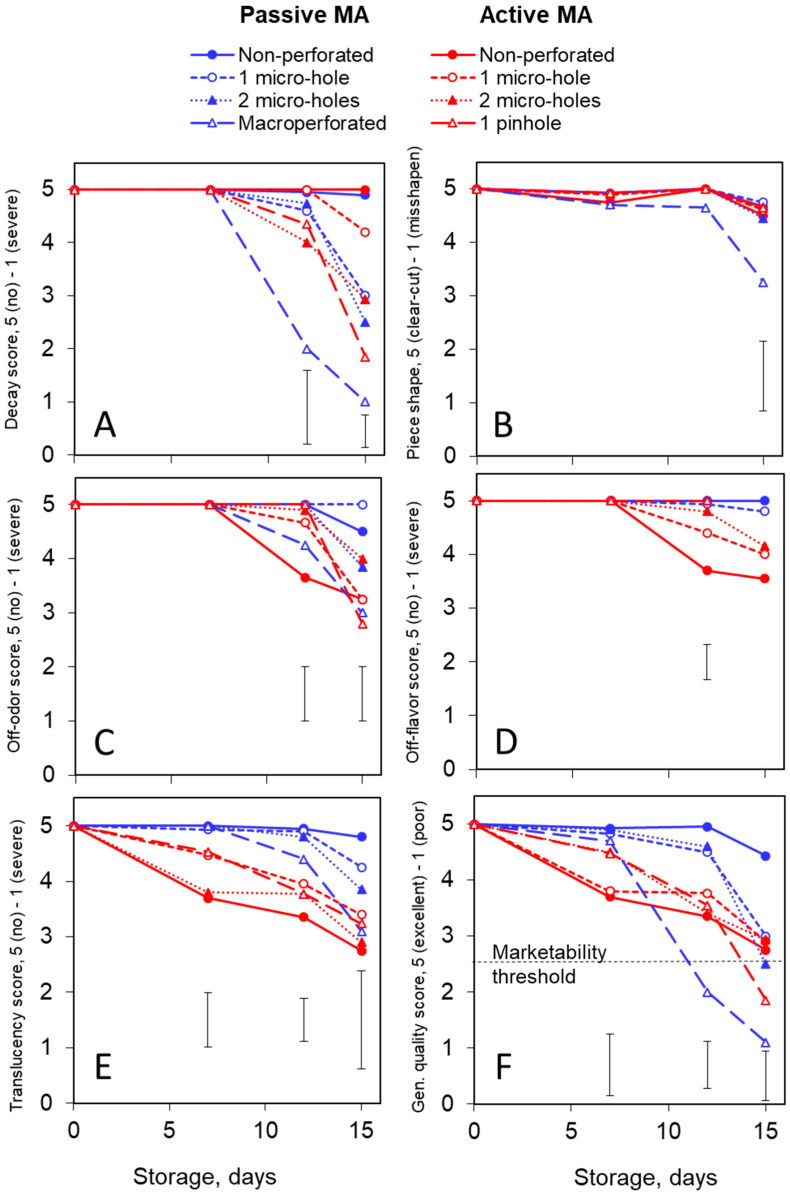
Effects of passive and active MA packaging on the quality scores of 100 g portions of fresh-cut *inodorus* melons stored for up to 15 days at 6–8 °C. Quality parameters: decay (**A**), piece shape (**B**), off-odor (**C**), off-flavor (**D**), translucency (**E**) and general quality/marketability (**F**). All quality parameters were evaluated according to a 5-grade visual scale, where a score of 5 corresponded to the highest quality, and a score of 1 to the lowest one. Individual data points are the means of three replications. Vertical bars represent honest significant differences (HSDs) (*p* ≤ 0.05) for each sampling period determined by the Tukey HSD test.

**Figure 4 foods-13-00256-f004:**
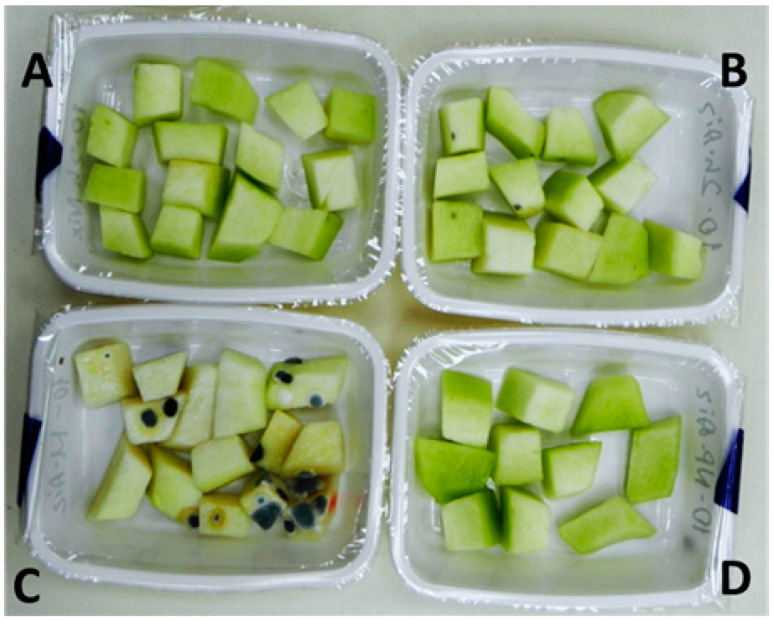
Effects of package perforation level on the appearance of 100 g portions of fresh-cut *cantalupensis* melons stored for 14 days at 6–8 °C. Perforation levels: a single micro-hole (ca. 70 µm) per package (**A**); two micro-holes (ca. 70 µm) per package (**B**); two macro-holes (2.5 mm) per package (**C**); and non-perforated package (**D**).

**Figure 5 foods-13-00256-f005:**
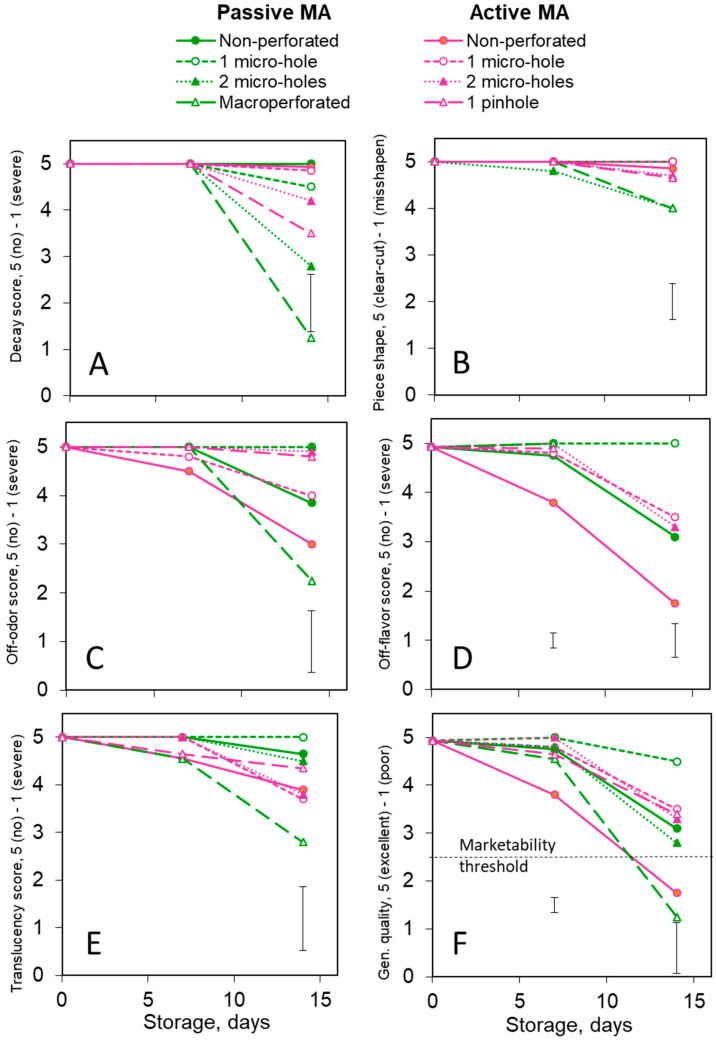
Effects of passive and active MA packaging on the quality scores of 100 g portions of fresh-cut *cantalupensis* melons stored for up to 14 days at 6–8 °C. Quality parameters: decay (**A**), piece shape (**B**), off-odor (**C**), off-flavor (**D**), translucency (**E**) and general quality/marketability (**F**). All quality parameters were evaluated according to a 5-grade visual scale, where a score of 5 corresponded to the highest quality, and a score of 1 to the lowest one. Individual data points are the means of three replications. Vertical bars represent honest significant differences (HSDs) (*p* ≤ 0.05) for each sampling period determined by the Tukey HSD test.

## Data Availability

Data is contained within the article or Appendix A.

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
