# Peer review of "Effects of Genotype and Modified Atmosphere Packaging on the Quality of Fresh-Cut Melons"

_foods, 2024, doi:10.3390/foods13020256_

Round 1

Reviewer 1 Report

Comments and Suggestions for Authors

This was an interesting study investigating the effects of modified atmosphere treatment on two different respiration types of fresh-cut melons. However, I have the following questions that need to be explained about the research content of this article:

The main purpose of this article involves the anaerobic respiration of fresh-cut melons in modified atmosphere packaging. However, the author only analyzed the gas components inside the package. So why did the author not measure the respiratory quotient of the melon or the cellular gas components inside the melon cells? This is important for The main purpose of this work is very important.

Author Response

Reviewer 1

Comments and Suggestions for Authors

This was an interesting study investigating the effects of modified atmosphere treatment on two different respiration types of fresh-cut melons. However, I have the following questions that need to be explained about the research content of this article:

The main purpose of this article involves the anaerobic respiration of fresh-cut melons in modified atmosphere packaging. However, the author only analyzed the gas components inside the package. So why did the author not measure the respiratory quotient of the melon or the cellular gas components inside the melon cells? This is important for The main purpose of this work is very important.

We are grateful to the reviewer for the positive evaluation of our study. The reviewer's valuable advice is interesting, primarily from the basic research viewpoint. However, our study was application-oriented, with the main purpose (see l. 78-80) of "investigating the performance of different MA packaging types as means to preserve the quality of fresh-cut melons… during simulated shelf life". Therefore, we put the emphasis on optimizing the package design for preventing the anaerobic fermentation, rather than on its in-depth investigation. For that purpose, we chose to measure the headspace level of volatile fermentation products as an informative marker of anaerobic processes. Following the reviewer's advice would demand a completely different research strategy and experimental plan.

Reviewer 2 Report

Comments and Suggestions for Authors

Dear authors

Thank you for the manuscript. I only have a few small additions.

Line 16,17, 28,..67, 72.…82-84..: C. melo var. cantalupensis and C. melo inodorus – please use italic. This has been taken into account from line 187

Line 29: If you write “on the other hand” you have to start with “on the one hand”

Line 66: the species include… – without s

Line 70: …and abscise (“slip”) at maturity. - what does this mean?

Line 71 – on the one hand – see above

Lime 102: 5% O2, 10% CO2 and 85% N2 – set numbers low

Line 106: Kibbutz

Line 121: electrochemical sensor via for O2

Line 122-124: The instrument’s needle was inserted in a package through an adhesive rubber 122 septum attached to the lidding film.  Sentence is repeated in lines 124-125

Line 176: Similar oxygen dynamics was were observed

Line 258: - on the other hand – see above

Line 346 and 353: on the other hand …

Line 430: At this point, it would be useful to point out that the cold chain must be maintained < 7°C for fresh cut melons to keep quality over such a long period.

Author Response

Reviewer 2

Dear authors, Thank you for the manuscript. I only have a few small additions.

Line 16,17, 28,..67, 72.…82-84..: C. melo var. cantalupensis and C. melo inodorus – please use italic. This has been taken into account from line 187

We are grateful to the reviewer for the attentive reading. The mistake in the nomenclature style occurred during the manuscript conversion to the journal's format. It has been re-checked and corrected throughout the manuscript.

Line 29: If you write “on the other hand” you have to start with “on the one hand”.

We thank the reviewer for the rigorous text examination. The sentence has been re-phrased. At the same time, rechecking the use of the idiom "on the other hand" with trustful sources has revealed that it is not strictly necessary to precede it with the phrase "on the one hand", see https://dictionary.cambridge.org/us/dictionary/english/on-the-other-hand https://www.merriam-webster.com/dictionary/on%20the%20other%20hand  

Line 66: the species include… – without s

The word "species" can be used both as singular and as plural; see for example https://dictionary.cambridge.org/dictionary/english/species . In line 66, we used this word as singular, addressing one species, Cucumis melo, and therefore used the "s" ending.     

Line 70: …and abscise (“slip”) at maturity. - what does this mean?

The horticultural term "slip" means that a cantalupensis-type melon can easily detach from the vine when it is fully mature. The explanation has been added to the text.

Line 71 – on the one hand – see above. Thank you, edited.

Lime 102: 5% O2, 10% CO2 and 85% N2 – set numbers low. Thank you, corrected.

Line 106: Kibbutz. Thank you, corrected.

Line 121: electrochemical sensor via for O2 Thank you, corrected.

Line 122-124: The instrument’s needle was inserted in a package through an adhesive rubber 122 septum attached to the lidding film.  Sentence is repeated in lines 124-125.

Thank you, edited.

Line 176: Similar oxygen dynamics was were observed. Thank you, corrected.

Line 258: - on the other hand – see above. Thank you, edited.

Line 346 and 353: on the other hand … Thank you, edited.

Line 430: At this point, it would be useful to point out that the cold chain must be maintained < 7°C for fresh cut melons to keep quality over such a long period.

We agree with the reviewer about the importance of cold chain conditions; a relevant statement has been added to the manuscript (l. 437). At the same time, our results were obtained at 7±1°C. Therefore, we did not stress the need of keeping the storage temperature below 7°C.

Reviewer 3 Report

Comments and Suggestions for Authors

Dear author,

Thanks for your good report. Please,

1- Add some recent related papers regarding MAP to the introduction and discussion sections. Most of the cited papers are not new.

2-  Line 88: for ca. 2 days? ca. = ?

3- Line 101-102: CO2 and O2. Please use subscript for 2.

4- Regarding different packaging I should mention that among all treatments (a) non-perforated, (b) having one laser micro-hole of ca. 70 μm per tray, or (c) two micro-holes, Only non-perforated can be considered as active MAP.  Treatments b, c, and d are not active MAP as you made small or large holes in packages. As you showed in Figure 1, O2 increased and CO2 decreased significantly in a, b, and c treatments after 5 days. It means they were not effective and suitable active MAP treatments. So, please change the name of these b, c, and d treatments to different packaging in your paper, and do not use the term active MAP.  This is also for passive MAP where you make micro holes in packages. 

5- As you mentioned in Figures 3 and 5, non-perforated packages had lower decay mainly due to higher CO2 inside packages during storage.

Regards

Comments on the Quality of English Language

It is OK.

Author Response

Reviewer 3

Dear author,

Thanks for your good report. Thank you for the positive evaluation of our work.

Please,

1- Add some recent related papers regarding MAP to the introduction and discussion sections. Most of the cited papers are not new.

Thanks for your advice, we have added some relevant up-to-date references (e.g., ## 19, 20, 34, 42, 45) substituting where possible the outdated ones.

2-  Line 88: for ca. 2 days? ca. = ? ca. = "circa", i.e. "around"; substituted with "about".

3- Line 101-102: CO2 and O2. Please use subscript for 2. Thank you, corrected.

4- Regarding different packaging I should mention that among all treatments (a) non-perforated, (b) having one laser micro-hole of ca. 70 μm per tray, or (c) two micro-holes, Only non-perforated can be considered as active MAP.  Treatments b, c, and d are not active MAP as you made small or large holes in packages. As you showed in Figure 1, O2 increased and CO2 decreased significantly in a, b, and c treatments after 5 days. It means they were not effective and suitable active MAP treatments. So, please change the name of these b, c, and d treatments to different packaging in your paper, and do not use the term active MAP.  This is also for passive MAP where you make micro holes in packages.

We thank the reviewer for the advice. The distinction between passive and active MA is determined by the initial package filling (air vs. gas mix, respectively) rather than by the packaging material. Excessive permeability, like in macro-perforated package, cancels any MA accumulation, no matter if the package is initially filled with air (passive MA) or a gas mix (active MA). However, minor package perforation (e.g. single micro-hole in our packages) allowed the MA generation. In this case, the effect of active vs. passive MA was significant during the initial transient period. For example, in the active MA with a single micro-hole the CO2 concentration was stable on the level of 8-9 kPa throughout the storage period (Fig. 1 C&D), while in the passive MA the CO2 accumulation started from 0 and reached the steady state (7-8 kPa) only by the end of the first week. Thus, the combination of active MA (gas flushing) with package micro-perforation did not prevent the MA formation and even stabilized its composition.

5- As you mentioned in Figures 3 and 5, non-perforated packages had lower decay mainly due to higher CO2 inside packages during storage.

Thank you for the advice. We have stressed this effect in the Discussion, see l. 348-349.

Reviewer 4 Report

Comments and Suggestions for Authors

This manuscript investigated the effects of variety and packaging method on the quality and sensory of melon fruit.

The product quality was evaluated by only three expert panelists. The results are therefore not sufficiently reliable. More panelists are required for the sensory evaluation.

SSC and firmness results should be provided and compared with the sensory evaluation results.

Author Response

Reviewer 4

This manuscript investigated the effects of variety and packaging method on the quality and sensory of melon fruit.

The product quality was evaluated by only three expert panelists. The results are therefore not sufficiently reliable. More panelists are required for the sensory evaluation.

In this study, we used the scoring method of sensory evaluation (Van Oirschot and Tomlins, 2002). According to the recommendations, the following number of assessors is required for this method: experts - one or more, trained assessors - 5 or more, untrained assessors - 20 or more. The panel comprised three expert panelists with many years of experience in the area, including an industry representative, and applied the evaluation template provided by the industry. Therefore, we are confident that such team was sufficient for performing the evaluation.

SSC and firmness results should be provided and compared with the sensory evaluation results.

We are grateful to the reviewer for this advice. In the revised version, the SSC and firmness measurement results have been presented in the supplementary materials. No significant differences between the treatments in SSC and firmness levels were revealed. The results of sensory evaluation were affected by hypoxic fermentation and microbial development rather than by the SSC level.

Round 2

Reviewer 3 Report

Comments and Suggestions for Authors

Dear author,

Thanks for the revised manuscript. However, the discussion did not change significantly. Also, I disagree with the authors respond regarding question 4.

Regards

Author Response

Reviewer:

Dear author, thanks for the revised manuscript. However, the discussion did not change significantly. Also, I disagree with the authors respond regarding question 4.

Answer:

We appreciate the reviewer's comment. The reviewer's opinion is that the term "active MAP" is inapplicable to perforated (including micro-perforated) packaging. Furthermore, the last sentence of the reviewer's comment #4, ("This is also for passive MAP where you make micro holes in packages") apparently means that he/she generally rejects the use of the term "MAP" to micro-perforated packaging. As recommended, we have extended the discussion, explaining the terms "modified atmosphere packaging (MAP)" and "active MAP" and justifying the use of these terms to micro-perforated packaging. The criterion for defining a packaging system as MAP is "alteration of the initial gaseous environment that surrounds the food so that the resulting environment affects the metabolic processes of the food and food-borne microorganisms" (McMillin, 2020). In some cases, as stressed by the reviewer, package perforation can indeed preclude the MA formation, making the in-package atmosphere identical to surrounding air. This may happen when (a) the food product is non-respiring or (b) the holes are too big or numerous. On the contrary, with respiring produce, optimized micro-perforated MAP is a well-established and commercially used method of fruit and vegetable preservation. The MA composition is determined by the balance between food's respiration and gas diffusion through the package, no matter if the initial atmosphere composition is air (so-called "passive MAP") or gas mix ("active MAP"). The initial atmosphere does not change the final steady-state MA composition but affects time duration needed to reach it. Combining micro-perforated packaging with "active MAP" sounds somewhat paradoxical (filling a package with a gas mix and at the same time letting it to escape through perforations) but in fact may help stabilizing the respiration-based MAP. This concept is an innovative element of the present manuscript. In the discussion, we have added several references to the publications implementing this concept. We hope that our argumentation may be accepted by the reviewer and/or by the scientific editor.

Reviewer 4 Report

Comments and Suggestions for Authors

The author(s) revised the manuscript.

Author Response

Rewiewer:

The author(s) revised the manuscript.

Answer:

We are grateful to the reviewer for accepting the revised version of the manuscript.